# How do various leadership practices affect professional learning communities? The mediating role of principals' perceived trust by teachers

Xiaobo Gu[1], Zhihui Liu[2]*, Zhenyuan Hang[1,3]

**1** College of Road and Bridge, Zhejiang Institute of communications, Hangzhou, Zhejiang, China,
**2** College of Education, Zhejiang University, Hangzhou, Zhejiang, China, **3** Department of Civil Engineering, Shantou University, Shantou, Guangdong, China

* 22203003@zju.edu.cn

## Abstract

Drawing on social exchange theory, this study aims to explore the effects of various leadership practices on professional learning communities (PLCs) and the mediating role of principals' perceived trust by teachers in the relationships between various leadership practices and PLCs from Chinese principals' perspective. Survey data were collected from 739 principals from different provinces. To examine the proposed model, the study utilized four-step hierarchical regression, Shapley value decomposition, and bootstrap methods. The results indicated that all the four components of leadership practices, namely setting directions, developing people, redesigning the organization and managing the instructional programme significantly and positively affected PLCs, and their contribution rates were 15.81%, 23.43%, 36.48% and 23.25% respectively. Principals' perceived trust by teachers was a significant mediator between all the four components of leadership practices and PLCs. The practical implications of the findings and suggestions for future research are discussed.

## Introduction

There has been a growing interest in the concept and research of professional learning communities (PLCs) over the past two decades. Accumulating evidence suggests that the development and sustainment of PLCs have a beneficial effect on school reform [1], teacher professional development [2,3], and student achievement [3,4]. Consequently, PLCs have been widely advocated by policy makers and educational practitioners around the world [5]. Understanding how PLCs are fostered and developed in diverse settings, including China, can offer valuable insights for global educational reform efforts.

Research on the antecedents of PLCs has been growing in recent years, yet several issues remain to be elucidated. First, principal leadership has been found to

**Data availability statement:** In the context of our study, the data that underpin the findings are available upon request from the corresponding author. Due to privacy and ethical considerations, the data are not publicly accessible. Specifically, the data cannot be shared publicly because they contain potentially identifiable information about the participants, which could compromise their confidentiality. To address these concerns while still facilitating scientific transparency and collaboration, data are available from the Zhejiang University Institutional Data Access / Ethics Committee for researchers who meet the criteria for access to confidential data. Interested researchers can contact the committee via email at [jiazhang2015@zju.edu.cn] or through the postal address provided below: Zhejiang University Institutional Data Access / Ethics Committee.

**Funding:** This research was funded by the National Social Science Fund of China (grant number: 20CSH033). The funders had no role in the study design, data analysis, decision to publish, or preparation of the manuscript.

**Competing interests:** The authors have declared that no competing interests exist.

play a crucial role in creating and sustaining PLCs [4,6]. However, extant studies have mainly focused on the relationships between various types of principal leadership such as distributed leadership [7], transformational leadership and instructional leadership [8] and PLCs, and the effects of comprehensive leadership practices on PLCs have been under-explored. Understanding this relationship can enrich our knowledge of how principals adopt specific leadership practices to support PLCs [9].

Second, existing studies on the relationships between principal leadership and PLCs have mainly depended on teachers' perceptions (e.g., [4,8]), while the perspective of principals has been ignored to a large extent. As studies have repeatedly found that misalignment existed between principals' and teachers' perspectives regarding the practices of school leadership [10,11], it is important to explore the influence of principal leadership on PLCs from the perspective of principals.

Third, and most important, the underlying mechanisms through which principal leadership affects PLCs are given little attention, and there is a lack of a well-developed theoretical framework to explain this relationship. To gain valuable insights into the dynamic mechanisms of how principal leadership affects PLCs, this study introduces the social exchange theory [12] as the theoretical basis. According to social exchange theory, when a leader offers valuable support, resources or opportunities to his/her subordinates, the subordinates are likely to respond positively with increased commitment, loyalty, trust in the leaders and cooperation with colleagues, and vice versa [13]. This reciprocal relationship offers a theoretical framework for analyzing the impact of principal leadership on PLCs. Guided by social exchange theory, when principals provide support for teachers to enhance their professional ability, teachers are more likely to give back by showing increased trust in principals [9]. When principals feel more trusted by teachers, they are likely to enhance the reciprocal dynamics of resource exchange [5]. This enhanced exchange of resources serves to foster a mutually beneficial relationship between teachers and principals, resulting in that principals provide stronger support for collaboration among teachers to promote their professional development, thus further accelerating the development of PLCs [14]. Therefore, it is probable that principals' perceived trust by teachers plays a mediating role in the relationship between principal leadership and PLCs.

To fill in the above-mentioned research gaps, the current study aims to explore the effects of various leadership practices on PLCs and whether principals' perceived trust by teachers mediates these effects from Chinese principals' perspective. In China, teachers' collaborative learning practices have existed for more than seven decades, which were regarded as one of the key factors in explaining Shanghai students' excellent performance in the Program for International Student Assessment [15]. In recent years, an increasing number of studies have used the concept of PLCs to examine teachers' collaborative practices in China, and found that principal leadership mattered a lot to the sustainable development of PLCs [5,16]. However, how principal leadership affects PLCs have been under-researched. Therefore, guided by the social exchange theory, this study examines how and whether principals' leadership practices influence PLCs through principals' perceived trust by

teachers in Chinese context, which can enrich our knowledge of the influencing mechanism of principal leadership on PLCs based on Chinese experience. Specifically, two research questions are put forward: (1) Do various aspects of principals' leadership practices affect PLCs in Chinese schools? What are the relative contributions of various aspects of principals' leadership practices to PLCs? (2) Does principals' perceived trust by teachers mediate the effects of various leadership practices on PLCs in Chinese schools? By exploring these questions, the study not only enhances our understanding of PLCs within the Chinese educational context but also contributes to the broader global discourse on the role of principal leadership and trust in promoting the development of PLCs.

## Literature review

### PLCs in Chinese context

It is generally believed that PLCs create a favorable environment for educators' continuous and collaborative sharing, interrogation and improvement of their professional practice with the purpose of promoting student learning [17]. Researchers have conceptualized PLCs in various educational contexts and suggested five basic characteristics of PLCs: shared purpose [18], a collective focus on student learning [19], collaborative activity [1], deprivatized practice [20], and reflective dialogue [18]. This conceptualization has been widely applied in both Western contexts [4,21] and non-Western contexts such as China (e.g., Yin and Zheng [5]; Zhang et al. [22]).

Although the term of PLC originated in Western Settings, structured teacher collaboration has already been integrated into the daily work of Chinese teachers for more than half a century [2]. As early as in 1952, the Ministry of Education of China required schools to set up Teaching Research Groups (jiaoyanzu, TGRs) among teachers of the same subject in order to improve teaching practice and education quality [23,24]. Various types of collaborative learning activities are regularly organized in TRGs, including joint lesson planning, collective inquiry into open lessons, collaborative action research, and so on [2].

Whether TRGs can be called PLCs has been debated. Unlike teachers' bottom-up initiative for working together as emphasized by PLC literature [25], TRGs are established according to the instruction of education authorities. For this reason, collective activities in TRGs are called "contrived collaboration" by some researchers (e.g., Wong [23]). Nevertheless, TRGs present the most basic features of PLCs, including collaborative learning, a collective focus on student learning, and shared practice [16]. Consequently, some researchers (e.g., Wang [26]) claimed that TRGs reflected "arranged genuine collegiality" instead of "contrived collegiality". Recent studies [2,26] have also repeatedly considered TRGs a typical form of PLCs in Chinese context. Following this perspective, we regard TRGs as a Chinese version of PLCs in this study.

An accumulating body of research has explored PLCs in Chinese context, such as the characteristics of PLCs in China [16], obstacles encountered in the development of PLCs [27], the influence of PLCs on teachers [28], and teacher learning in PLCs [29]. However, limited empirical studies, especially those from principals' perspective on the antecedents and their influencing mechanisms of PLCs have been conducted. Therefore, in this study, we explore the relationships among various leadership practices, principals' perceived trust by teachers and PLCs from principals' perspective.

### Leadership practices and PLCs

Extant studies have repeatedly shown that principal leadership is key to the practice of PLCs. For instance, Hassan et al.'s [14] study on teachers in Malaysia suggested that instructional leadership played a crucial role in shaping a favorable culture for PLCs. Torres' [7] multi-level analysis based on 2013 Teaching and Learning International Survey (TALIS) data showed that distributed leadership positively predicted teacher professional collaboration. Valckx et al.'s [8] study on 33 Flemish secondary schools indicated that transformational leadership had a positive effect on teachers' collective responsibility, which further positively affected their reflective dialogue.

It can be seen from the above that positive associations existed between various types of principal leadership such as distributed leadership, instructional leadership and transformational leadership and PLCs. Nevertheless, researchers

[30] claimed that there is a common core of leadership practices that should be grasped by all school leaders, regardless of which leadership model they adopt. Specifically, four types of leadership practices have been identified: (1) setting directions, referring to establishing a common goal among school members; (2) developing people, referring to providing personalized support and guidance to teachers and stimulating their intelligence; (3) redesigning the organization, referring to optimizing working conditions for teachers such as building collaborative cultures; and (4) managing the instructional programme, referring to supporting and monitoring teachers' teaching, protecting them from other distractions and promoting the improvement of teaching practices [30]. This four-dimension model of leadership practices has been found to suit the Chinese context [5,22]. Therefore, in the current study, we employ this model to define leadership practices.

Researchers have begun to pay attention to the relationships between leadership practices and PLCs. For example, Yin and Zheng's [5] study on Chinese teachers and Zhang et al.' [22] study on Chinese principals both found that leadership practices had a positive effect on all the components of PLCs. However, these studies have focused on the relationship between the whole construct of leadership practices and PLCs, and how various aspects of principals' leadership practices affect PLCs remains under-explored. Do various dimensions of leadership practices have significant effects on PLCs? And are the degrees of influence of various leadership practices dimensions different? Answering these under-researched questions is important, which can help enrich our knowledge of how principals adopt various leadership practices to facilitate PLCs.

## Leadership practices, principals' perceived trust by teachers and PLCs

Trust is one's confidence that others will fulfill their obligations in a reasonably predictable manner [31]. There are different kinds of trust in schools, such as trust among teachers, trust between teachers and principals and trust between teachers and students or parents [32]. Principals have been shown to play a key role in establishing a trust atmosphere in schools [33], which can not only promote teachers' innovative practices [9], but also contributes significantly to student achievement [34].

As trust, specifically trust among teachers and trust between teachers and principals, represents the quality of social relationships among school members, it is considered a key indicator of teacher collaboration [32] and a precondition for the development of PLCs [9]. Empirical studies have also reported the positive relationships between trust and PLCs. For instance, Gray et al.'s [35] survey suggested that collegial trust had a significant and positive effect on PLCs. Gray and Summers's [36] quantitative study also revealed that teachers' collegial trust and trust in principal contributed to PLC development.

Meanwhile, the mediating role of trust in the relationship between principal leadership and teacher professional learning or PLCs has been tested in several studies. For example, Li et al.'s [9] survey on 970 teachers from 32 primary schools in Hong Kong found that faculty trust served as a positive mediator between principal leadership and teacher professional learning. Similarly, Karacabey et al.'s [33] study on 1200 Turkish teachers showed that teacher trust positively mediated the effects of both principals' instructional and transformational leadership on teacher professional learning. However, Yin and Zheng's [5] survey on 1095 Chinese primary school teachers indicated that teachers' trust in colleagues positively, while their trust in principal negatively mediated the effect of leadership practices on the components of PLCs. Moreover, in Hallinger et al.'s [4] study on 559 teachers in 32 Hong Kong primary schools, the mediation of trust between principal leadership (including two dimensions, i.e., teaching and learning and professional development) and teacher professional community was not supported.

Thus, the studies that investigated whether trust mediated the relationship between principal leadership and PLCs have generated inconsistent results, which may be attributed to the different research sample, the different research contexts, or the different leadership variables used in the studies. Therefore, more empirical studies are needed to examine the mediating role of trust in the relationship between principal leadership and PLCs. Furthermore, most of the existing studies

have focused on the mediating role of trust among teachers, while research on the mediating role of trust between teachers and principals remains limited. And these studies have mainly depended on teachers' perceptions, and how principals perceive the role of trust in the outcome of their leadership has been under-explored, which needs further examination. According to social exchange theory [12], individuals would return the social benefits (e.g., trust, support, opportunities) they receive from others. Within the school context, when principals adopt the core leadership practices that successful leaders draw on, such as providing professional training and learning opportunities and establishing platforms for teacher sharing and collaboration, which function as supporting resources for teacher professional development [37], they are more likely to perceive a higher level of trust from their teachers based on the reciprocity principle of social exchange theory. In turn, a higher level of trust as perceived by principals would make them strengthen the mutually beneficial relationship by further supporting teachers' collaborative learning practices, thereby contributing to the advancement of PLCs [14]. Therefore, we speculate that principals' perceived trust by teachers serves as a mediator between principal leadership and PLCs.

In summary, the current study seeks to investigate the effects of various leadership practices on PLCs and their mediating relationships via principals' perceived trust by teachers. The conceptual framework is shown in Fig 1, in which various components of leadership practices are hypothesized to positively affect PLCs (H1), and principals' perceived trust by teachers is hypothesized to mediate the effects of various leadership practices on PLCs (H2).

## Method

### Participants

A comprehensive questionnaire survey was conducted among school principals from various provinces in China. Ethical approval for the study was obtained from the institutional review board prior to the survey's commencement, ensuring adherence to ethical standards. Recruitment was conducted from October 1, 2022 to January 30, 2023. All participants, who were adults, provided their informed consent in writing after being fully informed about the study's objectives, procedures, potential risks, benefits, and their rights, including the assurance of confidentiality and the option to withdraw from the study at any point without penalty. The written consent form served as a formal record of each participant's voluntary agreement to participate in the research, thus documenting and witnessing the consent process in accordance with the ethical guidelines. In total, nine hundred copies of the questionnaire were distributed, and 739 valid samples were obtained, giving a useful return rate of 82.11%. Of the participants, 65.6% (485) were male

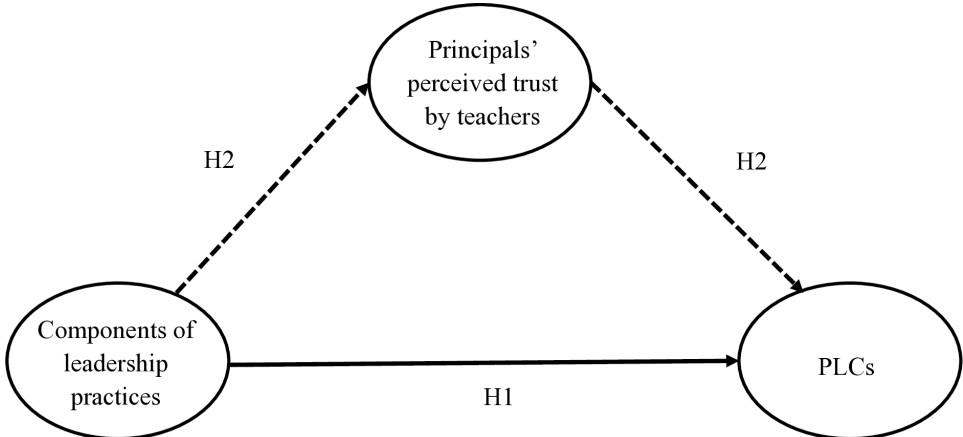

**Fig 1. Conceptual framework.** Note. Dotted line shows the indirect effect.

and 34.4% (254) were female. 63.7% (471) were principals and the other 36.3% (268) were deputy principals. 12.99% (96) of the principals aged at 40 years old or younger, 64.82% (479) were between 41–50 years old, and the remaining 22.19% (164) were 51 years old or older. There were 58.3% (431) of the participants working in elementary schools, 22.5% (166) in middle schools, 7.4% (55) in high schools, and 11.77% (87) in other types of schools such as nine-year education schools. 43.98% (325) of the principals worked in urban schools, and 56.02% (414) worked in suburban and rural schools.

## Instruments

The questionnaire comprised three scales, namely the leadership practice (LP) scale, the principals' perceived trust by teachers (PPTT) scale, and the professional learning community (PLC) scale. For all the three scales, the principals were requested to rate each of the item on a five-point Likert-type scale ranging from "strongly disagree" indicated by "1" to 5 "strongly agree" indicated by "5". In order to make the participants better understand the items, the questionnaire was developed in Chinese, i.e., the mother tongue of the participants.

The 21-item LP scale was adapted from Day et al. [30], containing 21 strategies adopted by principals to promote school improvement over the past decades. The scale comprises four sub-scales: setting directions (four items), developing people (five items), redesigning the organization (six items) and managing the instructional program (six items). A sample item was "I give my staff a sense of overall purpose". This scale has been used in Chinese context, such as Yin and Zheng's [5] study and showed good reliability and validity.

The 18-item PLC scale adapted from Leithwood, Aitken and Jantzi [38] comprises five sub-scales: shared sense of purpose (three items), collaborative activity (five items), collective focus on student learning (four items), deprivatised practice (three items) and reflective dialogue (two items). A sample item was"teachers in my school discuss teaching practices and behaviours of team members".This scale has been used in Chinese context, such as Zheng et al.'s [26]study and exhibited good reliability and validity.

The 5-item PPTT scale was adapted from Louis and Kruse [39]. A sample item was "I sense that teachers in my school understand the initial purpose and intention behind my work". This scale has been previously utilized in Zhang and Liu's [40] study on Chinese principals, demonstrating good reliability and validity.

In addition, considering the possible influence of principals' demographic variables on the relationships among variables of interest, principals' gender (1 = male, 2 = female), age (1 >= 51; 2 = 41–50; 3 =< 40), position (1 = elementary schools; 2 = middle schools; 3 = high schools; 4 = other schools), school level (1 = elementary schools; 2 = middle schools; 3 = high schools; 4 = other schools), geographic location (1 = urban areas; 2 = suburban and rural areas) and school type (1 = model school; 2 = ordinary school) were included in the questionnaire as the control variables. These control variables were transformed into dummy variables, and then were added into the data analysis.

## Analysis

SPSS 22.0, Amos 24.0 and Stata16.0 were used to analyze the data. First, confirmatory factor analysis (CFA) was conducted using Amos to examine the construct validity of the three scales. A number of standard indices were used to indicate the goodness of fit, including the chi-square statistic ($\chi2$), the root mean square error of approximation (RMSEA), the comparative fit index (CFI) and the Tracker-Lewis index (TLI). The fit of a model was considered good (or acceptable) if RMSEA ≤ 0.06 (0.08), CFI ≥ 0.95 (0.90), and TLI ≥ 0.95 (0.90) [41]. Meanwhile, convergent validity (average variance extracted value, AVE), internal consistency reliability (Cronbach's alpha coefficient), and composite reliability (CR) were also examined. Given that the self-reported data for this study came from the same participants, common method variance (CMV) may be an issue. To test whether there was the CMV problem, a latent variable model with a first-order factor containing all of the measurements as indicators was compared with the model without it [42]. Then, the descriptive statistics and correlations of all variables were calculated by SPSS.

After that, in order to examine both the direct and indirect effects of leadership practices on PLCs through principals' perceived trust by teachers, we adopted Kenny and Baron's [43] four-step hierarchical regression method using Stata. To avoid the multicollinearity problem in the multiple linear regression model, the variance inflation factor (VIF) test was carried out. If the VIF values of all variables were less than 10, we thought there was no multicollinearity in the model [44]. In addition, this study adopted the Shapley value decomposition method proposed by Shorrocks [45] to compare the degree of influence of independent variables on dependent variables. Last, bootstrap was used for the mediation analysis [46].

## Results

### Reliability and validity of the scales

The CFA results showed that the LP scale ($\chi^2 = 999.58$, df = 178, p < 0.001, RMSEA = 0.079, CFI = 0.94, TLI = 0.92), PLC scale ($\chi^2 = 515.57$, df = 109, p < 0.001, RMSEA = 0.07, CFI = 0.97, TLI = 0.96) and PPTT scale ($\chi2 = 12.82$, df = 4, p < 0.05, RMSEA = 0.05, CFI = 0.99, TLI = 0.99) all had satisfactory data fits. Table 1 shows that the Cronbach's alpha coefficients of the six variables ranged between 0.85 and 0.91, which were all higher than 0.70; the AVE values (0.67–0.72) were all higher than 0.50; and the CR values (0.91–0.93) were all higher than 0.70. Therefore, the three scales had good construct validity, convergent validity and reliability.

For the CMV test, we compared the measurement models with and without the CMV factor, and the results showed that the change of fitting index was not statistically significant according to Cheung and Rensvold's (2002) criterion: $\Delta\chi^2$/df = 0.06 (less than 0.1), $\Delta$CFI = -0.006 (less than 0.1), $\Delta$TLI = 0.002 (less than 0.1), $\Delta$RMSEA = 0.01 (less than 0.1). Therefore, CMV is unlikely to have greatly distorted the interpretation of the findings.

### Descriptive statistics and correlations

The results of descriptive statistics and correlations are also shown in Table 1. For the four subscales of the leadership practices, the mean scores ranged from 4.30 to 4.60, which were relatively high. Specifically, setting directions had the highest score (M = 4.60, SD = 0.52), followed by managing the instructional programme (M = 4.43, SD = 0.58) and redesigning the organization (M = 4.43, SD = 0.57). Developing people had the lowest score (M = 4.30, SD = 0.62). The mean scores of principals' perceived trust by teachers (M = 4.58, SD = 0.54) and PLCs (M = 4.36, SD = 0.56) were also relatively high.

**Table 1. Descriptive statistics, correlation matrix, Cronbach's α, AVE, CR, and square root of AVE.**

|  | SD | DP | RO | MIP | PPTT | PLCs |
|---|---|---|---|---|---|---|
| SD | 1 |  |  |  |  |  |
| DP | 0.58** | 1 |  |  |  |  |
| RO | 0.58** | 0.65** | 1 |  |  |  |
| MIP | 0.55** | 0.59** | 0.64** | 1 |  |  |
| PPTT | 0.51** | 0.56** | 0.57** | 0.62** | 1 |  |
| PLCs | 0.53** | 0.60** | 0.68** | 0.58** | 0.59** | 1 |
| M | 4.61 | 4.30 | 4.43 | 4.43 | 4.58 | 4.36 |
| SD | 0.52 | 0.62 | 0.57 | 0.58 | 0.54 | 0.56 |
| α | 0.87 | 0.85 | 0.90 | 0.91 | 0.89 | 0.90 |
| AVE | 0.72 | 0.68 | 0.68 | 0.70 | 0.72 | 0.67 |
| CR | 0.91 | 0.91 | 0.93 | 0.93 | 0.93 | 0.91 |

Note:

**p < 0.01, SD = setting directions, DP = developing people, RO = restructuring the organization, MIP = managing the instructional program, PPTT = principals' perceived trust by teachers, PLCs = professional learning communities.

Furthermore, significant and positive correlations existed among the six variables, and the correlation coefficients ranged between 0.51 and 0.68.

## Regression analyses and Shapley value decomposition

The regression analysis and Shapley value decomposition were carried out to examine the direct and indirect effects of leadership practices on PLCs through principals' perceived trust by teachers, and the results are presented in Table 2. Before the regression analysis, we conducted the VIF test, and the results showed that the VIF values of the independent variables ranged from 1.06 to 4.54, which were all less than 10, thus there was no multicollinearity in the model.

**Table 2. Regression analyses and Shapley value decomposition.**

| Variables | $R^2$ | B | SE | shaply(%) |
|---|---|---|---|---|
| Step 1: Model 1 | 0.788 | | | |
| SD | | 0.09*** | 0.03 | 15.81% |
| DP | | 0.16*** | 0.03 | 23.43% |
| RO | | 0.47*** | 0.04 | 36.48% |
| MIP | | 0.19*** | 0.03 | 23.25% |
| Control variables | | YES | YES | 1.04% |
| Dependent variable: PLCs | | | | |
| Variables | $R^2$ | B | SE | shaply(%) |
| Step 2: Model 2 | 0.805 | | | |
| SD | | 0.08*** | 0.02 | 12.07% |
| DP | | 0.12*** | 0.03 | 18.31% |
| RO | | 0.45*** | 0.03 | 32.84% |
| MIP | | 0.05* | 0.04 | 16.21% |
| PPTT | | 0.27*** | 0.03 | 19.44% |
| Control variables | | YES | YES | 1.12% |
| Dependent variable: PLCs | | | | |
| Variables | $R^2$ | B | SE | shaply(%) |
| Step 3: Model 3 | 0.726 | | | |
| SD | | 0.05* | 0.03 | 13.41% |
| DP | | 0.18*** | 0.04 | 21.76% |
| RO | | 0.08* | 0.04 | 20.86% |
| MIP | | 0.54*** | 0.04 | 42.63% |
| Control variables | | YES | YES | 1.38% |
| Dependent variable: PPTT | | | | |
| Variables | $R^2$ | B | SE | shaply(%) |
| Step 4: Model 4 | 0.637 | | | |
| PPTT | | 0.84*** | 0.02 | 97.48% |
| Control variables | | YES | YES | 2.52% |
| Dependent variable: PLCs | | | | |

Note:

*$p < 0.05$,

***$p < 0.001$, SD = setting directions, DP = developing people, RO = restructuring the organization, MIP = managing the instructional program, PPTT = principals' perceived trust by teachers, PLCs = professional learning communities.

Table 2 shows that in the first regression analysis, all the four leadership practice components (independent variables) affected PLCs (dependent variable) in a positive and significant way, and they accounted for 78.80% of the total variance of PLCs. Specifically, restructuring the organization had the greatest influence ($\beta = 0.47$; $p < 0.001$), whose contribution rate was 36.48%. The next was developing people ($\beta = 0.16$; $p < 0.001$) and managing the instructional programme ($\beta = 0.19$; $p < 0.001$), whose contribution rate was 23.43% and 23.25% respectively. Setting directions had the least influence ($\beta = 0.09$; $p < 0.001$), whose contribution rate was 15.81%. Therefore, H1 was supported.

Principals' perceived trust by teachers (mediation variable) was added to the second regression model. The result showed that the five variables explained a larger proportion (80.50%) of the total variance of PLCs (dependent variable). Among them, principals' perceived trust by teachers ($\beta = 0.27$; $p < 0.001$) affected PLCs in a positive and significant way, and the contribution rate was 19.44%. And the effect of the other four variables, i.e., setting direction, developing people, restructuring the organization and managing the instructional programme were similar to those in the first regression model, although the $\beta$ coefficients and contribution rates fluctuated a little.

The third regression analysis result showed that the four leadership practice components (independent variables) explained 72.60% of the total variance of principals' perceived trust by teachers (mediation variable). Among them, all of four components, i.e., managing the instructional programme ($\beta = 0.54$; $p < 0.001$), developing people ($\beta = 0.18$; $p < 0.001$), restructuring the organization ($\beta = 0.08$; $p < 0.050$) and setting directions ($\beta = 0.05$; $p < 0.050$) (independent variables) affected principals' perceived trust by teachers (mediation variable) in a positive and significant way, and their contribution rates were 42.63%, 21.76%, 20.86% and 13.41% respectively.

The last regression analysis result showed that principals' perceived trust by teachers (mediation variable) affected PLCs (dependent variable) in a positive and significant way ($\beta = 0.84$; $p < 0.001$), explaining 63.70% of the total variance of PLCs. The contribution rate of principals' perceived trust by teachers for the variance of PLCs was 97.48%.

The above results indicated that principals' perceived trust by teachers was very likely a mediator between leadership practices and PLCs. Nonetheless, the mediating relationships resulted from regression should be further tested by additional mediation analyses [9]. Therefore, we further employed bootstrap analysis to demonstrate the statistical significance of the mediating effects.

## Mediation analysis

A mediation analysis was conducted based on 1000 bootstrapping samples to examine the mediating effects of principals' perceived trust by teachers on the relationships between leadership practice components and PLCs. The results are shown in Table 3, in which the standardized estimates of indirect effects within the 95% confidence interval (CI) is reported.

According to Hayes [46], an indirect effect is significant if zero does not locate between the upper and lower boundaries of the 95% CI. The results showed that principals' perceived trust by teachers was a significant mediator between all the

Table 3. Mediation analysis of principals' perceived trust by teachers on the relationships between leadership practice components and PLCs.

| Independent variables | Dependent variable | Mediation variable | Estimate (SE) | 95% CI | P |
|---|---|---|---|---|---|
| SD | PLCs | Principals' perceived trust by teachers | 0.33 (0.04) | [0.26, 0.40] | 0.000 |
| DP | | | 0.31(0.03) | [0.25, 0.38] | 0.000 |
| RO | | | 0.24(0.03) | [0.19, 0.29] | 0.000 |
| MIP | | | 0.36(0.04) | [0.29, 0.44] | 0.000 |

Note: SD = setting directions, DP = developing people, RO = restructuring the organization, MIP = managing the instructional program, PPTT = principals' perceived trust by teachers, PLCs = professional learning communities.

four leadership practice components (i.e., setting directions, developing people, restructuring the organization, and managing the instructional program) and PLCs. Therefore, H2 was supported.

## Conclusion and discussion

Drawing on social exchange theory, this study explored the effects of various leadership practices on PLCs and the mediating role of principals' perceived trust by teachers in the relationships between leadership practices and PLCs from Chinese principals' perspective. Results indicated that all the four components of leadership practices, namely setting directions, developing people, redesigning the organization and managing the instructional programme had significant and positive effects on PLCs in Chinese schools, and their contribution rates were 15.81%, 23.43%, 36.48% and 23.25% respectively. Principals' perceived trust by teachers significantly mediated the effects of all the four components of leadership practices on PLCs. The findings enrich the international literature on the relationship between principal leadership and PLCs by offering valuable insights from the Chinese experience, contribute to the application of social exchange theory in explaining how leadership practices affect PLCs and introduce the noteworthy perspective of principals to research in this field. Importantly, these insights, grounded in the Chinese context, offer a potential foundation for understanding similar dynamics between leadership, trust, and PLCs development in diverse global educational settings.

### The effects of various components of leadership practices on PLCs

Consistent with the results obtained by previous studies from teachers' perspectives (e.g., Hallinger et al. [4]), this study found that leadership practices mattered significantly to PLCs, as they explained 78.80% of the variance of PLCs. Meanwhile, unlike previous studies (e.g., Yin & Zheng [5]) that focused on the relationship between the whole construct of leadership practices and PLCs, the current study explored the effects of various components of leadership practices on PLCs. The results showed that all the four components of leadership practices had significant effects on PLCs as perceived by Chinese principals, which provides further evidence for the critical role of leadership practices in the development of PLCs in Chinese context [22].

First, redesigning the organization had the largest influence on PLCs, indicating that it was the most important contributor to PLCs. This result lends credence to previous studies indicating that when principals provide teachers with favorable working conditions, especially collaborative structures and cultures, teachers' collaborative learning and professional development can be greatly promoted [37]. The current study further suggests that restructuring schools into communities where leaders focus on relationship building and mutual sharing serves as the cornerstone of PLCs [34,47].

Second, the effects of developing people and managing the instructional programme on PLCs came next to that of redesigning the organization, and they had similar contribution rates. On the one hand, this result corroborates previous research showing that principals' leadership practices in terms of developing people such as providing individualized support and learning opportunities for teachers can enhance their professional competence [30,48]. It also highlights that principals focusing on learning for all school members and facilitating teachers' personal growth can drive the development of PLCs [4]. On the other hand, the current findings indicate that it is critically important for principals to put teaching and learning in the first place and provide necessary guidance and support for teachers to improve instructional practice [37]. While previous studies [27,49] show that if principals fail to protect their teachers from outside distractions such as heavy administrative duties, the development of PLCs would be hindered; the current study further suggests that principals focusing on high-quality teaching and learning can promote the practice of PLCs [14].

Third, although setting directions contributed the least to PLCs compared to the other three components of leadership practices, it produced a significant effect on PLCs. As shared vision and purpose is regarded an essential characteristic of PLCs [3,18], it is significant that principals help form a common sense of purpose among all school members, which can contribute to effective enactment of PLCs [6,49].

**The mediating role of principals' perceived trust by teachers in the effects of leadership practices on PLCs**

As trust is often considered the glue for teacher interactions in schools [31], it was posited as a mediator between principal leadership and PLCs in this study. Unlike previous studies mainly focusing on the mediating role of faculty trust from the perspective of teachers, the current study investigated the mediating effect of principals' perceived trust by teachers on the relationships between the various components of leadership practices and PLCs from principals' perspective. The results revealed that principals' perceived trust by teachers significantly and positively mediated the effects of all the components of leadership practices on PLCs. This finding is similar to previous studies indicating that trust served as a positive mediator between principal leadership and teacher professional learning [33,36]. It also echoes Tschannen-Moran' [50] proposition that trust, as a key indicator of social capital, is essential for PLC development.

However, the current finding is inconsistent with Hallinger et al.'s [4] research showing that there was no mediation of organizational trust (including both trust in colleagues and trust in principal) between principal leadership and PLCs. This may be because the research contexts are different. The significant mediating effect of principals' perceived trust by teachers in this study can be explained by the cultural characteristics of China. Specifically, the Chinese collectivist culture underlines maintaining social harmony and interpersonal relationships [26]. In this context, Chinese principals attach great importance to building harmonious interpersonal relationships, including positive relations between them and teachers [16]. When principals perceive a higher level of trust from teachers, they are more likely to exchange with teachers openly, deliver personalized assistance to improve teachers' teaching practices, and provide tailored support for teachers' professional learning [47,50], which can result in well-functioning PLCs.

Meanwhile, the finding of this study is also different from that of Yin and Zheng's [5] study suggesting that teacher's trust in principal negatively mediated the effect of leadership practices on PLC components according to Chinese teachers' perceptions. This may be attributed to the different perspectives adopted by the two studies. In teachers' view, they need to spend time and effort to build trust relationships with principals, and their interactions with principals may focus more on non-professional issues such as resource allocation, thus their trust in principal is helpless for PLC practice [5]. By contrast, principals tend to perceive that when teachers trust them, they can better support teachers' collaborative learning practices, thus contributing to the development of PLCs [14]. In other words, compared to teachers, principals take a more positive view of trust between them and teachers. In this sense, this study calls for more research, especially that from multiple perspectives, on the mediating role of trust between principals and teachers in the associations between principal leadership and PLCs.

## Implications

The present study has important implications for school leaders and policy makers, especially those in China, and suggests broader applications for educational settings globally.

First, it is necessary for leadership training projects to help principals strengthen their leadership practices. This study illustrated that all the four components of leadership practices mattered for both principals' perceived trust by teachers and PLCs. Thus, it is essential to help principals master the core leadership practices and apply them in daily work. Specifically, principals should grasp how to set shared vision and goals among faculty around school development, teacher professional growth and student learning outcomes [7], provide professional support for teachers to enhance their continuous learning and development [9], establish a facilitative organizational structure and favorable culture to foster teachers' collaborative practices [30,47], and create a working environment where teachers focus on the improvement and innovation of teaching practices [14].

Above all, due to that redesign the organization had the largest influence on PLCs, principals should pay special attention to optimizing teachers' working conditions. On one hand, principals need to provide adequate time, space and communication structure for teachers to collaborate, nurture a culture of sharing and collaboration in schools, and offer teachers learning opportunities and ample resources [22]. On the other hand, principals should create an inspiring school

environment by setting clear guidelines and implementing tangible projects to encourage meaningful learning, and establish positive and harmonious relationships with families and communities.

Second, trust between principals and teachers should be enhanced. This study indicated that principals' perceived trust by teachers positively mediated the relationships between the four types of leadership practices and PLCs. Therefore, both principals and teachers should attach importance to trust building and make efforts to cultivate trustworthy relationships [5,23]. Particularly for principals who take the primary responsibility for the tone of trust in schools [32], it is important for them to build an environment of trust and set an example for teachers in building trusting relationships [50]. Specifically, principals should be consistent in their words and actions, act as trusted leaders, show respect, understanding and care for teachers and students [34,51], interact with them on an equal footing [5], and work towards the school vision and goals through practical actions.

## Limitations and future recommendations

Several limitations were involved in this study. First, the sample is not completely random, which limits the generalizability of the research results. Future studies should adopt more rigorous sampling strategies such as stratified random sampling to verify the current findings. Second, this study only relies on principal's perceptions. Future research could further examine the relationships among leadership practices, trust and PLCs by combining and comparing teachers' and principals' perspectives, considering the differences between teachers' and principals' perceptions as reported in the current study and previous studies [5,11]. Third, this study only focuses on trust between principals and teachers, and future studies can investigate the mediating effects of both trust between principals and teachers and trust among teachers on the relationships between leadership practices and PLCs.

## Author contributions

**Conceptualization:** Zhihui Liu.

**Data curation:** Zhenyuan Hang.

**Formal analysis:** Zhihui Liu.

**Funding acquisition:** Zhihui Liu.

**Investigation:** Xiaobo Gu.

**Methodology:** Xiaobo Gu, Zhihui Liu, Zhenyuan Hang.

**Project administration:** Xiaobo Gu, Zhenyuan Hang.

**Resources:** Zhenyuan Hang.

**Writing – original draft:** Xiaobo Gu, Zhihui Liu.

**Writing – review & editing:** Zhihui Liu, Zhenyuan Hang.

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
