## [Decision Letter · Decision Letter 0]

13 Feb 2025

PONE-D-24-60309How do various leadership practices affect professional learning communities? The mediating role of principals’ perceived trust by teachersPLOS ONE

Dear Dr. Liu,

Thank you for submitting your manuscript to PLOS ONE. After careful consideration, we feel that it has merit but does not fully meet PLOS ONE’s publication criteria as it currently stands. Therefore, we invite you to submit a revised version of the manuscript that addresses the points raised during the review process.

**ACADEMIC EDITOR: Minor revisions**

We look forward to receiving your revised manuscript.

Kind regards,

Agbotiname Lucky Imoize

Academic Editor

PLOS ONE

Journal Requirements:

2. Thank you for stating the following financial disclosure: [This research was funded by the National Social Science Fund of China (grant number: 20CSH033)].

Please state what role the funders took in the study. If the funders had no role, please state: ""The funders had no role in study design, data collection and analysis, decision to publish, or preparation of the manuscript."

Additional Editor Comments:

Dear Authors,

Revise the paper according to the reviewers' comments and check the English for improvement.

Reviewers' comments:

Reviewer's Responses to Questions

**Comments to the Author**

1. Is the manuscript technically sound, and do the data support the conclusions?

Reviewer #1: Yes

Reviewer #2: Yes

2. Has the statistical analysis been performed appropriately and rigorously?

Reviewer #1: Yes

Reviewer #2: Yes

3. Have the authors made all data underlying the findings in their manuscript fully available?

Reviewer #1: Yes

Reviewer #2: Yes

4. Is the manuscript presented in an intelligible fashion and written in standard English?

Reviewer #1: Yes

Reviewer #2: Yes

5. Review Comments to the Author

Reviewer #1: The manuscript is methodologically rigorous, addresses a relevant topic, and provides original contributions. While focused on the Chinese context, it meets the standards for publication. Emphasising global implications could further strengthen the discussion.

Reviewer #2: Overall, I find this study to be well-designed and and clearly presented. The research addressed important gaps in the literature and provided valuable insights into the mechanism through which leadership practices influence professional learning communities. Below are my detailed comments and recommendations.

Strengths of the Manuscript

1. Significance and Novelty: The study examined the influence of various leadership practices on PLCs and the mediating role of principals' perceived trust by teachers. This is an important area of research, given the increasing emphasis on PLCs in educational reform and the limited existing research on the mediating role of trust from the principal's perspective. The use of social exchange theory provided a suitable framework for understanding the relationship between leadership practices and PLCs.

2. Methodological Rigor: The study employed a quantitative research method, with a large sample size and robust statistical techniques, including four-step hierarchical regression, Shapley value decomposition, and bootstrap methods. These methods ensure the reliability and validity of the findings. The inclusion of CFA to validate constructs and the assessment of CMV demonstrate thoroughness in addressing potential biases.

3. Clear Presentation: The manuscript was well-organized with a clear structure that guided the reader through the research process. Each section, from the abstract to the discussion, is logically presented. The findings were clearly articulated, and the study's implications were discussed in a way that provided useful recommendations for practitioners and policymakers.

Minor Issues and Suggestions

1. Some references were relatively old. I suggest the authors to add more recent studies, particularly those related to the relationships between leadership practices and PLCs, to enhance the timeliness of the research.

2. The description of control variables was brief. I suggest the authors to provide more detailed information about these variables. For example, what specific positions were included under "position"? What levels of schools were categorized under "school level"? Clarifying these details will help readers better understand the scope and precision of the study's control mechanisms.

6. PLOS authors have the option to publish the peer review history of their article (what does this mean? ). If published, this will include your full peer review and any attached files.

**Do you want your identity to be public for this peer review?** For information about this choice, including consent withdrawal, please see our Privacy Policy .

Reviewer #1: No

Reviewer #2: No

---

## [Author Response · Author response to Decision Letter 1]

7 Mar 2025

A detailed response to the reviewers' comments has also been submitted in document form, which provides a comprehensive overview of the changes made and addresses each point raised by the reviewers.

Response to Reviewers

Dear Editors,

We wish to thank the anonymous reviewers for their careful reading of the manuscript and helpful comments. We believe we have used these comments in straightforward way to improve the research. Our detailed responses to the review comments are presented in a point by point format. Related changes are highlighted in red font in the main text.

Reviewer 1 :

Comments to the Author

The manuscript is methodologically rigorous, addresses a relevant topic, and provides original contributions. While focused on the Chinese context, it meets the standards for publication. Emphasising global implications could further strengthen the discussion.

Response: Thank you for the positive comments. We are pleased that our manuscript has been recognized for its methodological rigor and original contributions. We have taken your suggestion to heart and have revised the manuscript to better emphasize its global implications. Specifically, we have made the following changes in the “Introduction” “Conclusion and Discussion” and “Implications” sections, and have also added relevant international literature to support our views and findings. These revisions are aimed at enhancing the international impact of our study. (please see page 3/5/20/23).

Introduction Section:

We have added the following sentences to highlight the global relevance of our study:

"Understanding how PLCs are fostered and developed in diverse settings, including China, can offer valuable insights for global educational reform efforts." (please see page 3).

"By exploring these questions, the study not only enhances our understanding of PLCs within the Chinese educational context but also contributes to the broader global discourse on the role of principal leadership and trust in promoting the development of PLCs." (please see page 5).

These additions emphasize that our research on PLCs in China can provide broader insights applicable to educational reforms worldwide.

Conclusion and Discussion Section:

We have expanded the discussion on the theoretical contributions of our study:

"The findings enrich the international literature on the relationship between principal leadership and PLCs by offering valuable insights from the Chinese experience, contribute to the application of social exchange theory in explaining how leadership practices affect PLCs and introduce the noteworthy perspective of principals to research in this field. Importantly, these insights, grounded in the Chinese context, offer a potential foundation for understanding similar dynamics between leadership, trust, and PLCs development in diverse global educational settings." (please see page 20).

This integration of Chinese educational practices with global research underscores the broader applicability of our findings.

Implications Section:

We have emphasized the broader applications of our study:

"The present study has important implications for school leaders and policy makers, especially those in China, and suggests broader applications for educational settings globally." (please see page 23).

By highlighting the broader applications, we aim to show that our research can inform educational practices beyond China.

Reviewer 2 :

Comments to the Author

Overall, I find this study to be well-designed and and clearly presented. The research addressed important gaps in the literature and provided valuable insights into the mechanism through which leadership practices influence professional learning communities. This study is well-designed and clearly presented, addressing important gaps in the literature on how leadership practices influence PLCs through the mediating role of teachers' trust in principals. The use of social exchange theory and robust quantitative methods, including hierarchical regression and validation techniques, ensures the reliability and validity of the findings. The manuscript is logically structured, with clear articulation of findings and practical implications. Overall, this research makes a valuable contribution to the field.

Response: Thank you for the positive comments.

Minor Suggestions: Some references were relatively old. I suggest the authors to add more recent studies, particularly those related to the relationships between leadership practices and PLCs, to enhance the timeliness of the research.

Response: Thank you for your suggestion. We have carefully reviewed our references and added several studies published in the past five years to enhance the timeliness and relevance of our research. The corresponding citations in the main text have been updated and highlighted in red to clearly indicate these changes.

Minor Suggestions: The description of control variables was brief. I suggest the authors provide more detailed information about these variables. For example, what specific positions were included under "position"? What levels of schools were categorized under "school level"? Clarifying these details will help readers better understand the scope and precision of the study's control mechanisms.

Response: Thank you for your suggestion. We have expanded the description of the control variables in the "Method" section to provide more clarity. The revised text is as follows:

“Principals' gender (1 = male, 2 = female), age (1 = ≥51; 2 = 41–50; 3 = ≤40), position (1 = elementary schools; 2 = middle schools; 3 = high schools; 4 = other schools), school level (1 = elementary schools; 2 = middle schools; 3 = high schools; 4 = other schools), geographic location (1 = urban areas; 2 = suburban and rural areas), and school type (1 = model school; 2 = ordinary school) were included in the questionnaire as control variables. These control variables were transformed into dummy variables and then added to the data analysis.” (please see page 13).

Please include a copy of Table 3 which you refer to in your text on page 18.

Response: Thank you for your attention to our manuscript. We have reviewed the content and found that the table caption "Mediation analysis of principals' perceived trust by teachers on the relationships between leadership practice components and PLCs" was incorrectly labeled as Table 2. Following your guidance, we have now corrected this to Table 3.

---

## [Decision Letter · Decision Letter 1]

1 Apr 2025

How do various leadership practices affect professional learning communities? The mediating role of principals’ perceived trust by teachers

PONE-D-24-60309R1

Dear Dr. Liu,

We’re pleased to inform you that your manuscript has been judged scientifically suitable for publication and will be formally accepted for publication once it meets all outstanding technical requirements.

Kind regards,

Agbotiname Lucky Imoize

Academic Editor

PLOS ONE

Additional Editor Comments (optional):

The manuscript is acceptable in its current form.

Reviewers' comments:

Reviewer's Responses to Questions

**Comments to the Author**

1. If the authors have adequately addressed your comments raised in a previous round of review and you feel that this manuscript is now acceptable for publication, you may indicate that here to bypass the “Comments to the Author” section, enter your conflict of interest statement in the “Confidential to Editor” section, and submit your "Accept" recommendation.

Reviewer #2: All comments have been addressed

2. Is the manuscript technically sound, and do the data support the conclusions?

Reviewer #2: Yes

3. Has the statistical analysis been performed appropriately and rigorously?

Reviewer #2: Yes

4. Have the authors made all data underlying the findings in their manuscript fully available?

Reviewer #2: Yes

5. Is the manuscript presented in an intelligible fashion and written in standard English?

Reviewer #2: Yes

6. Review Comments to the Author

Reviewer #2: The authors have adequately addressed my comments raised previously. I have no other comments. I think the paper can be published in its current form.

7. PLOS authors have the option to publish the peer review history of their article (what does this mean? ). If published, this will include your full peer review and any attached files.

**Do you want your identity to be public for this peer review?** For information about this choice, including consent withdrawal, please see our Privacy Policy .

Reviewer #2: No

---

## [Editor Report · Acceptance letter]

PONE-D-24-60309R1

PLOS ONE

Dear Dr. Liu,

I'm pleased to inform you that your manuscript has been deemed suitable for publication in PLOS ONE. Congratulations! Your manuscript is now being handed over to our production team.

Kind regards,

on behalf of

Mr. Agbotiname Lucky Imoize

Academic Editor

PLOS ONE